# Molecular basis of the lipid-induced MucA-MucB dissociation in *Pseudomonas aeruginosa*

Tao Li[1], Lihui He[1], Changcheng Li[1], Mei Kang[2], Yingjie Song[1], Yibo Zhu[1], Yalin Shen[1], Ninglin Zhao[1], Chang Zhao[1], Jing Yang[1], Qin Huang[1], Xingyu Mou[1], Aiping Tong [1], Jinliang Yang[1], Zhenling Wang[1], Chengjie Ji[3], Hong Li[1], Hong Tang[1] & Rui Bao [1]✉

MucA and MucB are critical negative modulators of sigma factor AlgU and regulate the mucoid conversion of *Pseudomonas aeruginosa*. Previous studies have revealed that lipid signals antagonize MucA-MucB binding. Here we report the crystal structure of MucB in complex with the periplasmic domain of MucA and polyethylene glycol (PEG), which unveiled an intermediate state preceding the MucA-MucB dissociation. Based on the biochemical experiments, the aliphatic side chain with a polar group was found to be of primary importance for inducing MucA cleavage. These results provide evidence that the hydrophobic cavity of MucB is a primary site for sensing lipid molecules and illustrates the detailed control of conformational switching within MucA-MucB in response to lipophilic effectors.

[1] Center of Infectious Diseases, State Key Laboratory of Biotherapy and Cancer Center, West China Hospital, Sichuan University and Collaborative Innovation Center of Biotherapy, Chengdu, China. [2] Department of Laboratory Medicine, West China Hospital, Sichuan University, Chengdu, China. [3] Clinical Laboratory of Sichuan Academy of Medical Science & Sichuan Provincial People's Hospital, Chengdu, China. ✉email: baorui@scu.edu.cn

To adapt to changing environments, Gram-negative bacteria usually use the envelope stress responses (ESR) system to coordinate gene expression[1]. In pathogenic bacteria, ESR controls numerous cellular processes such as, virulence factor production, motility, antibiotic resistance, and bacterial survival[1,2]. In most ESR systems, extra-cytoplasmic function sigma factors ($\sigma^{ECF}$) are crucial transcription elements required for stress signal transmission and amplification[3]. $\sigma^{ECF}$ is specifically inhibited by the cognate anti-sigma factors, whereas the relief of this inhibition is frequently regulated by a cascade of cleavage reactions named regulated intramembrane proteolysis (RIP)[4].

AlgU (also known as AlgT or $\sigma^{22}$) is the key $\sigma^{ECF}$ of *P. aeruginosa*. It is responsible for transduction of the extracellular stimulus that regulates mucoid transition of *P. aeruginosa* in cystic fibrosis (CF) patients[5]. MucA and MucB are cognate anti-sigma factors for AlgU activity[5]. In non-mucoid strains, MucB forms a stable complex with MucA that serves as a fine-tune control mechanism that protects MucA from cleavage. It has been proven that misfolded outer-membrane proteins (OMPs) can activate periplasmic protease AlgW and thus initiate the proteolysis of MucA by cleaving its periplasmic domain[6]. Increasing evidences indicate that the major signals for MucA proteolysis require both unfolded OMPs and off-pathway lipopolysaccharides (LPS) induced by extracellular stress[7,8].

Because of the physiological importance of *mucA* and *mucB*, their mutations are commonly associated with a persistent and mucoid phenotype of *P. aeruginosa* that has been isolated from the lungs of cryptogenic fibrosing alveolitis (CFA) patients[9]. Recently, a structural studies of MucB, MucA$^{peri}$–MucB, and MucA$^{cyto}$–AlgU revealed a redox-sensitive stress–response mechanism in MucB[10]. However, despite intense investigations, there remain many unanswered questions about MucA/MucB-associated RIP signaling. Of interest is determining how signal molecules such as LPS influence the MucA–MucB complex and release the MucA for AlgW degradation. In this study, we provide structural evidence that the hydrophobic core-occupied MucB undergoes dramatic conformational changes in the regions of residues 92–113, 192–216 and 230–237, exposing the cleavage site of MucA to the solvent. We also show that the fatty acid moiety is crucial for inducing the release of MucA from MucB. Using site-directed mutagenesis and a functional assay, we verified the roles of the critical residues involved in the MucA–MucB interaction and MucB–lipid association, providing experimental support to interpret the mechanism of the *mucA/mucB*-controlled mucoid phenotype in *P. aeruginosa*.

## Results

### MucB exclusively protects MucA$^{peri}$ from AlgW degradation.
AlgW, MucA, and MucB are the functional equivalents to DegS, RseA, and RseB from *Escherichia coli* and all sense and transmit outer-membrane stress through similar mechanisms[7,8,11]. AlgW and DegS both belong to the PDZ-containing serine proteases and possess high sequence identity (42.5%), but different substrate specificities[7]. To evaluate the function and exchangeability of the Muc- and Rse-systems, we overproduced and purified the periplasmic parts of MucB/RseB and AlgW/DegS (without the signal peptide and transmembrane domain) to reconstitute the proteolysis events of the periplasmic regions of MucA (MucA$^{peri}$: residues 106–194) and RseA (RseA$^{peri}$: residues 120–216). Both MucB and RseB were tested in two sets of protease cleavage experiments in the presence of a peptide agonist (YVF)[7]. MucB and RseB worked well to prevent cleavage in their parental systems but could not be substituted for each other, suggesting that the requirement for the strict choice of functional elements might

lie in the compatibility of MucA/RseA and MucB/RseB (Supplementary Fig. 1A). Next, we systematically investigated the profile of the MucB-controlled AlgW cleavage toward MucA. As Supplementary Fig. 1B shows, the protective effect of MucB on MucA was not persistent and decreased after prolonged incubation. Nonetheless, a certain amount of MucA was intact after 1 h of reaction. By comparison, MucB suppressed AlgW action in a ratio-dependent manner in that full protection of MucA required at least an equal molar equivalent of MucB.

To further investigate the specific recognition between MucA$^{peri}$ and MucB, we measured their binding affinity ($K_d$ value of 1.5 μM) by using isothermal titration calorimetry (ITC) and successfully obtained the MucA$^{peri}$–MucB complex in a 1:1 ratio via a Ni–NTA column and subsequent gel-filtration purification (Supplementary Fig. 2A, B). The modest interaction but moderate selectivity and stability of the MucA–MucB complex might be indicative of the exquisite control over the RIP under diverse stress conditions.

### Overall structure of MucA$^{peri}$–MucB.
The MucA$^{peri}$–MucB was crystallized and the structure was solved by molecular replacement (MR) using the N-terminal domain (NTD: residues 25–208) and C-terminal domain (CTD: residue 209–315) of *E. coli* RseB as the MucB model. Based on the electron density map generated from the MR, we were able to build residues 145–191 of MucA. The final structure was refined to 1.9 Å, with $R_{work} = 0.1729$ and $R_{free} = 0.1929$ (Supplementary Table 1).

MucB is composed of an NTD (residues 22–210) and smaller CTD (residues 211–313) (Fig. 1). The MucB-NTD is characterized by a half β-barrel fold with ten antiparallel β-strands (βA-βJ) and an α-helix (α1). The hydrophobic inner side faces the C-terminal domain, which is composed of six-stranded (βL, βM, βN, βO, βP, βQ) and two-stranded (βK, βR) twisted antiparallel β-sheets and a helix (α6). The concave sheets from the two domains present an open and accessible cavity to accommodate the periplasmic domain of MucA. In agreement with the RseA$^{peri}$–RseB interactions (Fig. 1), the MucA helix element αI (residues R157–S177) acted as the major binding element during MucA–MucB coupling[12,13].

The RseA$^{peri}$–RseB complex (PDB code: 3M4W, the sequence identity between MucB and RseB, MucA and RseA are 30.16% and 34.15%, respectively) revealed that the RseA$^{peri}$$_{142-151}$ is buried in the hydrophobic pocket of RseB, and thus avoids cleavage by DegS[13]. In our MucA$^{peri}$–MucB complex structure, the N terminal of MucA$^{peri}$ (residues 106–145) is too flexible to be detected in the crystal structure (Fig. 1). Remarkably, the MucA$^{peri}$–MucB complex structure revealed two extra binding regions. First, MucA$_{145-158}$, corresponding to the structure-unsolved RseA$_{159-168}$ segment, exhibited an αβ turn conformation and formed antiparallel main-chain hydrogen bonds with the βO of MucB. Second, in contrast with the invisible RseA$_{190-216}$, MucA$_{181-191}$ adopted a helical conformation (αII) and bound to MucB-NTD. As was observed for the primary binding in αI, the similar extra binding patterns observed in MucA–MucB may also exist in the Rse system and we speculate that the MucA$^{peri}$–MucB complex structure represents an intermediate state (inter-state) that is different from the fully protected state (protect-state) presented in the RseA$^{peri}$–RseB complex[13].

### The conformational gating of MucB and release of MucA.
Both the apo and RseA$^{peri}$-bound forms of RseB have been solved[12–14], and both show that the spatial arrangement of the N- and C-terminal domains is rigid. Such structural rigidity is attributed to the extensive inter-domain interactions (Supplementary Fig. 3), including either direct hydrogen bonding or hydrophobic

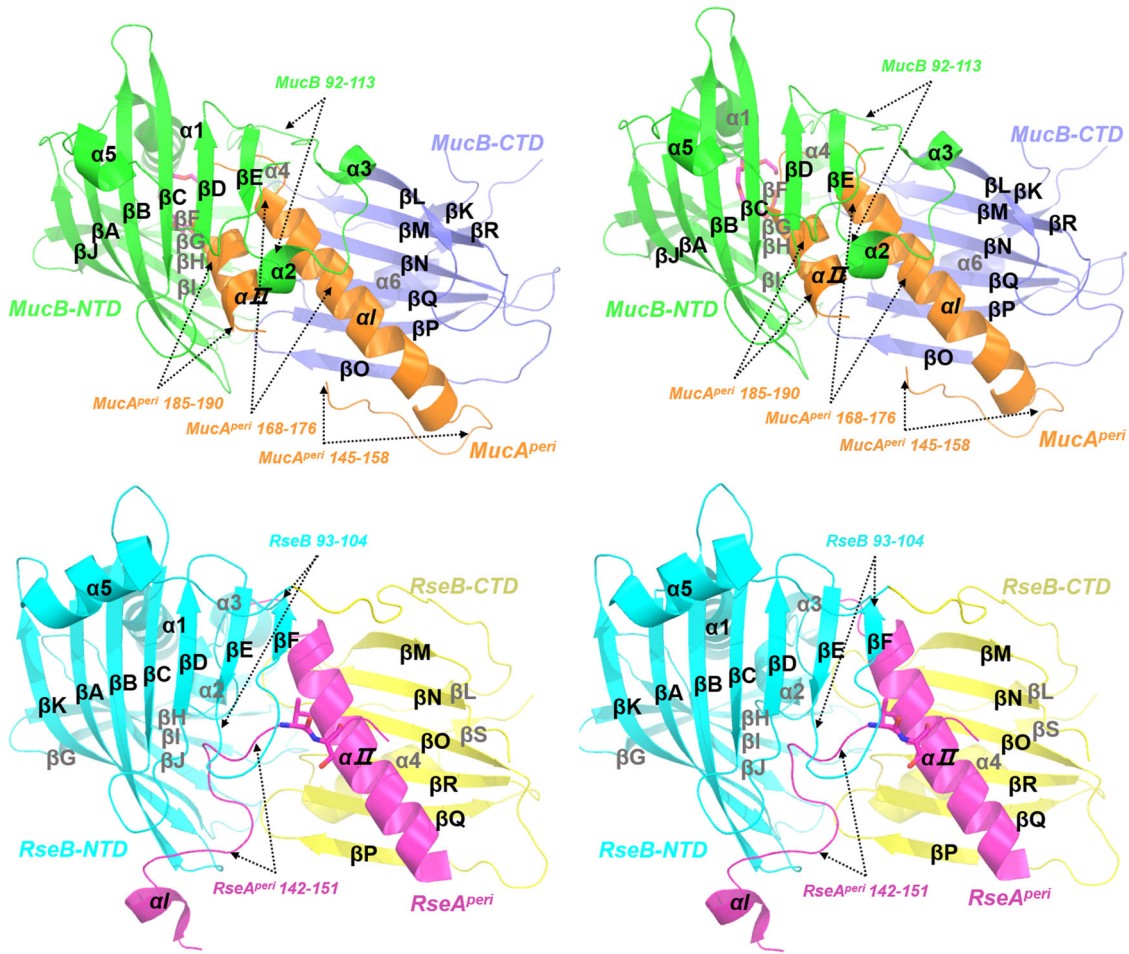

**Fig. 1 The side-by-side overall stereo structure views of the MucA^peri–MucB and RseA^peri–RseB complexes.** MucAperi–MucB complex (top) and the RseAperi–RseB complex (PDB code: 3M4W, bottom). The MucA^peri is colored in orange, the NTD, CTD of MucB are colored in green and blue. A bound PEG molecule is shown as hot-pink sticks (top). Accordingly, the RseA^peri is colored in magenta, the NTD, CTD of RseB are colored in cyan and yellow, respectively. These figures were generated by PyMOL (http://www.pymol.org).

interactions represented by a buried inter-domain surface area of 925.9 Å$^2$. In MucB, the central residues such as, Y119, R146, Y147, and D247 are structurally conserved with those of RseB. However, the inter-domain interface area is slightly smaller (847.1 Å$^2$) (Supplementary Fig. 3). Structural superposition of the MucA^peri-bound MucB with the RseA^peri-bound RseB gave a rmsd (root mean square deviation) value of 1.85 Å for 239 equivalent Cα (Fig. 2a). The major conformational variations are at three loop regions including residues 92–113, 192–216, and 230–237.

Comparing RseA^peri–RseB and MucA^peri–MucB (Fig. 1 and 2a), the MucA^peri αII inserted into the inner cavity of the MucB-NTD and a highly conserved proline P112 (corresponding to P112 in RseB) underwent a large dihedral angle transition (φ = −92.1 ψ = −4.6 in MucB compared to φ = −62.3 ψ = 149.8 in RseB) that allowed MucB$_{92–113}$ to flip to the surface. Consequently, MucB$_{91–104}$ (corresponding to RseB$_{93–104}$), which forms the 6th β strand in RseB and protects the N terminal of RseA^peri$_{142–151}$, adopted a helical conformation (α3) and interacted with MucA^peri$_{168–176}$ and MucA^peri$_{185–190}$ (Fig. 1). These distinct secondary structure elements may associate with structural variations between MucB$_{192–216}$ and RseB$_{200–224}$, although MucB had a unique disulfide bond (C90–C198) connecting the N-terminal region of the loop and βE. In addition, MucA^peri bound closer to the hydrophobic side of the MucB-NTD (Fig. 2b).

By comparing with the recently reported MucB-apo (6IN8) structure[10], we found that the main conformational variations occurred in the C-terminal domain and the loop region containing residues 92–113 (Fig. 2c). In MucB-apo structure, the loop$_{92–113}$ folded into the cavity of MucB, while in MucA-complexed form, MucB$_{92–113}$ shifted outward to generate adequate space for MucA binding (Fig. 2c). Another proline residue P106 was found to be involved in this structural transition. This rearrangement allowed MucA to form αII, and reduced the interactions between the C and N domains of MucB. Therefore, MucA binding initiates the MucB$_{92–113}$ rearrangement and MucA αII anchoring, leading MucA N-terminal cleavage site to be exposed to the solvent. This conversion might facilitate the subsequent AlgW coupling.

**Interactions between MucA^peri and MucB.** The periplasmic domain of membrane-spanning anti-sigma factors from different species were aligned together based on the structure superposition between MucA^peri and RseA^peri (Fig. 3a). The selected homologs were divided into Rse- and Muc-type groups. The sites that were observed or predicted to be involved in hydrogen and charge–charge interactions were aligned as shown in Fig. 3a, b. The primary binding between MucA/RseA αI and the MucB/RseB C-terminal domain represent a universal determinant for anti-sigma factors coupling because the RseA^peri truncation with αI alone still interacts with RseB[12].

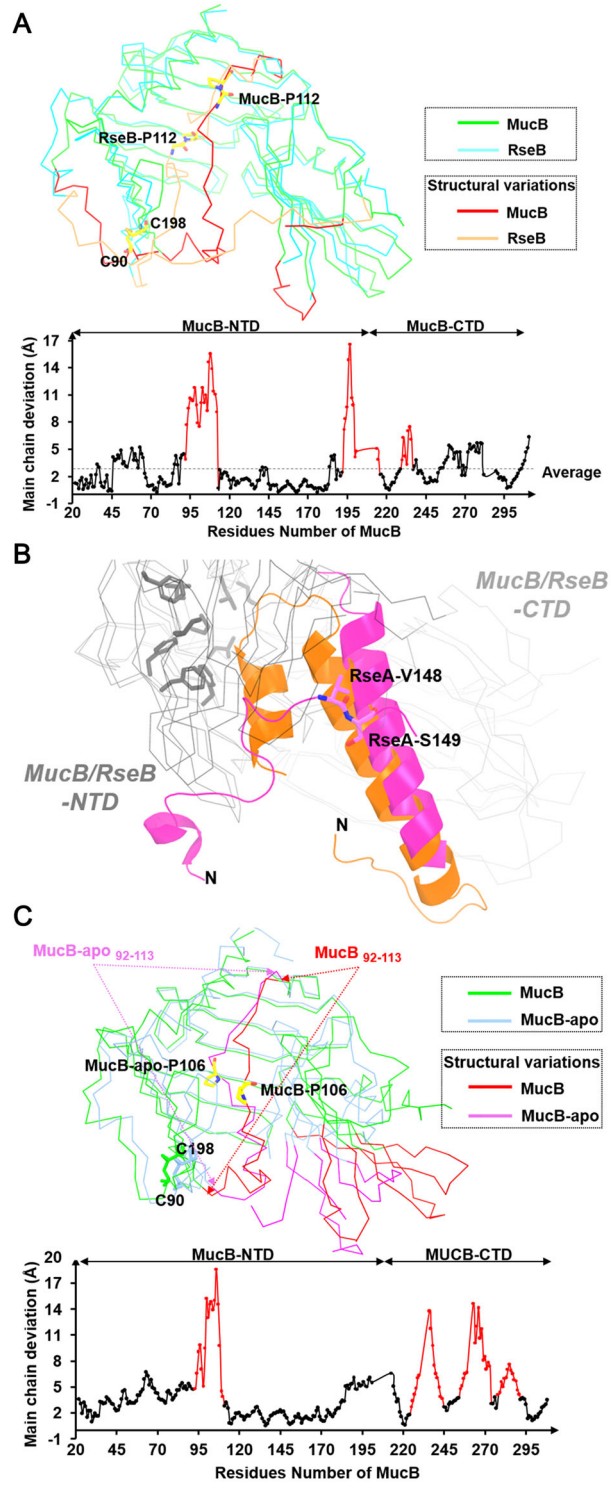

**Fig. 2 Structural comparisons of MucB and RseB. a** Overview of superimposed structures (top) and the Cα RMSD (root mean square deviation) plots (bottom) of MucB and RseB. The loop regions (residues 92–113, residues 192–216, and residues 230–237 in MucB) that generate most structural variations between MucB and RseB are shown in red and orange. Conserved proline residues are shown as yellow sticks. **b** Superimposed structure of MucA[peri]–MucB and RseA[peri]–RseB. MucA[peri], RseA[peri] are shown as orange and magenta cartoon. The site (V148/S149) of RseA[peri] degradation by DegS was shown with sticks and colored in magenta. The NTD and CTD of MucB/RseB are displayed in gray ribbon. The hydrophobic amino acids (L28/L31/F40/F44/I57/L151/F176/F178) in the hydrophobic side of MucB are displayed in gray sticks. **c** Overview of superimposed structures (top) and the Cα RMSD (root mean square deviation) plots (bottom) of MucB and MucB-apo (6IN8). The loop regions (residues 92–113, residues 225–245, residues 247–275, and residues 279–291 in MucB) that generate most structural variations between MucB and MucB-apo are shown in red and magenta. Proline-106 (P106) is displayed with yellow sticks.

that the αII region can be deleted without losing the MucA[peri]–MucB association, also increased the structural flexibility and facilitated the conformational changes that mediated the transition from protect-state to inter-state.

**The binding pocket for the lipid-A moiety of LPS in MucB.** MucB and RseB can function as periplasmic LPS sensors, recognizing and interacting with the lipid-A portion[8]. The MucB-NTD forms a U-shape beta-half-barrel and a large hydrophobic cavity with an approximate surface area of 3500 Å[2], making this domain the most likely to bind to LPS. The MucA[peri]–MucB complex structure had a polyethylene glycol (PEG) molecule bound, which may be incorporated during crystallization (Fig. 4a; Supplementary Fig. 4A). MucB[99–111] is too flexible to be modeled in recently reported MucA–MucB complex structure (6IN9) (Supplementary Fig. 4B)[10]. In our PEG-bound structure, MucB[92–113] is stable and can be modeled. The circular form of this PEG molecule, although different from the liner form of lipid or PEG molecules in other lipoproteins such as LolB[15], provides an opportunity to visualize the binding site for the acyl chains of lipid-A. Notably, most hydrophobic residues that line the PEG-binding pocket are conserved among MucB/RseB homologs (Fig. 3b and 4a), supporting the view that binding of "off-pathway" LPS molecules is a common feature in this family[8]. The limited diameter (13–17 Å) of this concave cavity suggests that MucB may bind a single lipid-A molecule at this site. In a recently reported protein structure in complex with LPS (6S8H)[16], lipid-A occupies a range of 5–19 Å, which confirms that the MucB cavity is able to accommodate part of the lipid-A moieties. The similar binding regions of PEG in MucA[peri]–MucB and the RseB[108–117] in RseA[peri]–RseB (Fig. 2a) indicates that the rearrangements of the MucB-NTD and MucB[92–113] regions may be associated with LPS binding (Fig. 2c).

The L-IIA unit of LPS has been identified as the minimal active fragment in MucA[peri]–MucB dissociation (Fig. 4b)[8], while the attachment of a single PEG group to MucB did not fully dissociate the MucA[peri]–MucB complex. This observation inspired us to investigate the effects of the substructures of lipid-A on MucA[peri] cleavage. Several types of molecules including lipid-A (Sigma), LPS (Solarbio) or boiled LPS, detergents (n-dodecyl-β-D-maltopyranoside, DDM; n-nonyl-β-D-glucopyranoside, NG; n-octyl-β-D-glucopyranoside, β-OG), polyethylene glycol monomethyl ethers of different molecular weights (550, 2000, 3350, and 5000 Da), organic solvents (DMSO; (+/−)-2-methyl-2,4-pentanediol,

To understand the contribution of residues that participate in MucA[peri]–MucB interactions (Fig. 3c), a variety of MucA mutants were generated and subjected to the His-affinity pull-down experiments (Fig. 3d). First, most of the MucA mutants still retained the ability to form complexes with MucB except for three residues in the αI region (W158, R162, and H170A), suggesting that the extensive interaction network may provide evolvability that underlies the anti-sigma factor recognition specificity. Second, the existence of redundant contacts, especially the fact

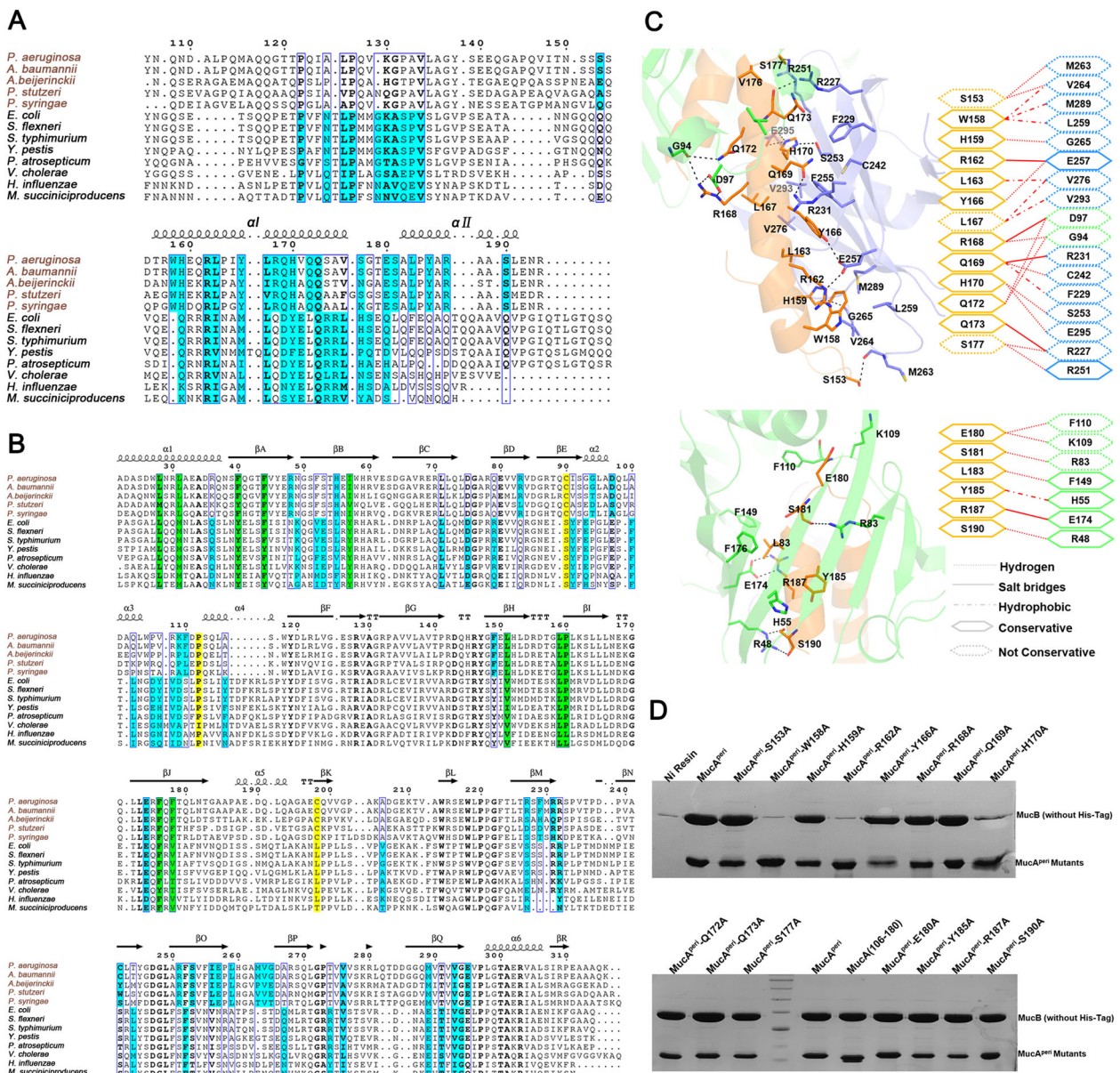

**Fig. 3 Structural-based sequence alignment and specific interactions in MucA^peri-MucB complex. a** Multi-sequence alignment (generated by Espript 3.0, http://espript.ibcp.fr/ESPript/ESPript/) of MucA/RseA homologous. Residues involved in protein–protein interactions are shaded in cyan. **b** Multi-sequence alignment of MucB/RseB homologous. Conserved hydrophobic residues that form lipid-binding pockets are shown in green, positions corresponding to residues P112, C90, and C198 in MucB (disulfide bond) are shaded in yellow. **c** Structures and schematics of the binding interfaces between MucA^peri (in orange) αI (top)/αII (bottom) and MucB (NTD, CTD are colored green and blue). **d** Pull-down assay. Incubate 20 μg MucA^peri or its variants (with His-tag) with excessive MucB (without His-tag), the molar ratio of MucA^peri to MucB was 1:3 to ensure excess of MucB interact with MucA^peri or mutants. Fractions were eluted with solution buffer containing 300 mM imidazole and determined by 15% SDS-PAGE gel and visualized by Coomassie Brilliant Blue stain.

MPD; isopropanol; glycerol), disaccharides (maltose, β-D-gluco-pyranosyl-D-glucose (β-DGDG)) and fatty acids with various chain lengths (C8–C16) were chosen and separately added to the AlgW-mediated MucA^peri proteolysis system. After a 30 min incubation at 37 °C, the samples were subjected to SDS-PAGE (Fig. 4b). Lipid-A and LPS exhibit positive effects to relief the AlgW cleavage inhibition caused by MucB, whereas boiled LPS under alkali solution which would generated activate fragment L-IIA could induce MucA degradation of the same magnitude as that induced by lipid-A[8] (Fig. 4b). Similar to lipid-A, detergents consisting of a saccharide moiety and a linear alkyl tail exhibited

significant effects on MucA^peri cleavage, and the detergent with longer acyl chain had more obvious effect (Supplementary Fig. 5A). Instead, MucA^peri remained intact when exposed to PEGs, organic solvents and disaccharides, while the fatty acids showed certain degrees of activation on MucA^peri degradation. These data suggest that an alkyl chain with a hydrophilic head group are both functional groups of lipid-A for inducing MucA^peri release. This speculation was supported by the subsequent experiment (Supplementary Fig. 5B) in which the lauric acid (C12)-induced MucA^peri cleavage process was obviously accelerated by adding disaccharides. In addition, a lower band (around 25 kDa) was observed

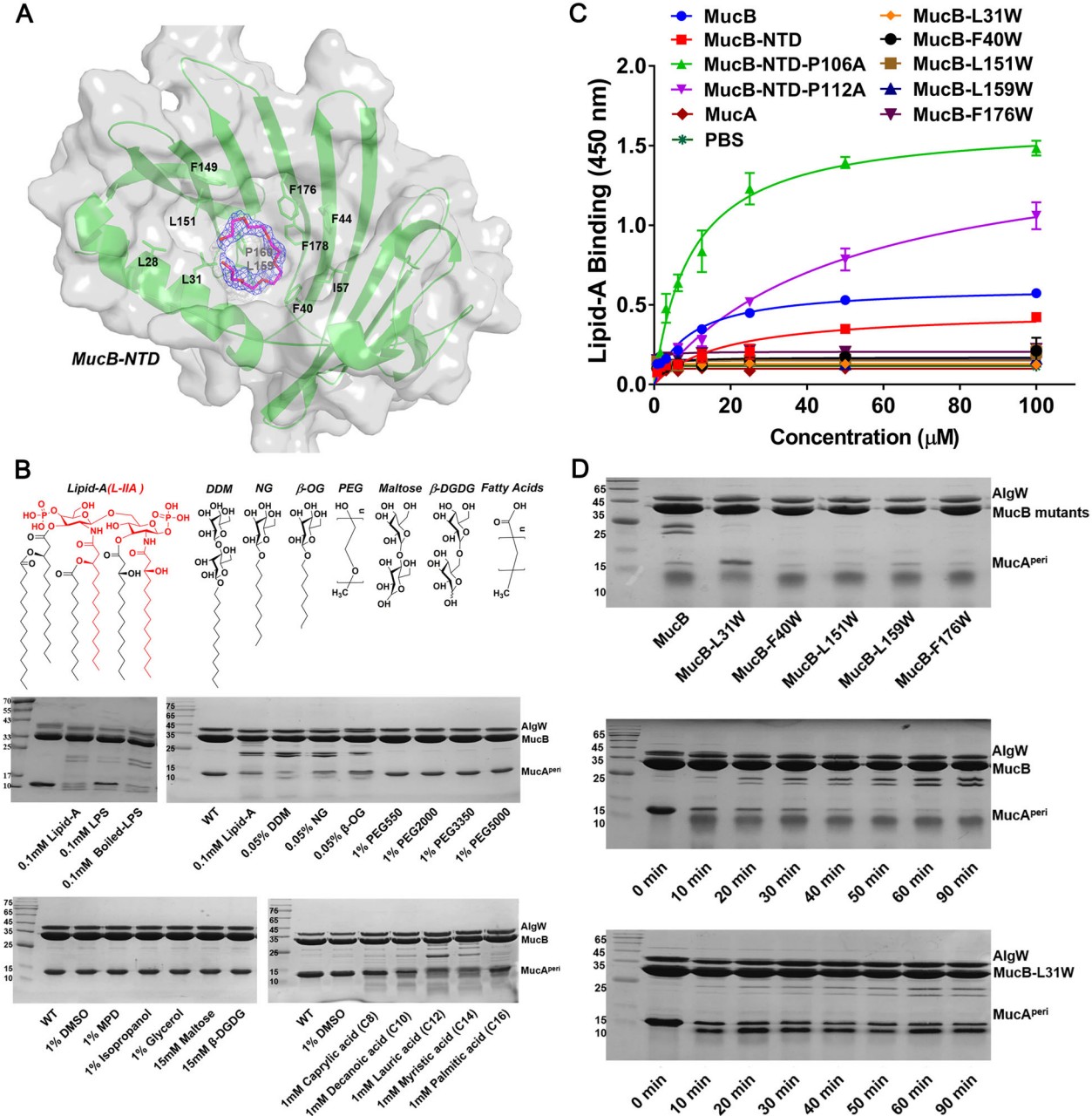

**Fig. 4 Structural and biochemical analysis of the lipid-binding pocket in MucB. a** A polyethylene glycol (PEG) molecular bound in the hydrophobic cavity of MucB structure. The hydrophobic residues surrounding PEG are shown with green sticks. The polyethylene glycol monomethyl ether 550 is shown in hot-pink stick and its 2mFo-DFc map (1.5σ) is displayed as blue mesh. **b** SDS-PAGE assay of MucA$^{peri}$ (125 μM) degradation by AlgW (25 μM) in the presences of MucB (130 μM), YVF peptide (80 μM), and different reagent including 0.1 mM lipid-A hydrolyzed with NaOH (L-IIA was colored in red), 0.1 mM LPS (Solarbio) or boiled LPS, n-dodecyl-β-D-maltopyranoside (0.05% DDM), n-nonyl-β-D-glucopyranoside (0.05% NG), n-octyl-β-D-glucopyranoside (0.05% β-OG), polyethylene glycol monomethyl ether reagents like 1% PEG550, 1% PEG2000, 1% PEG3350, 1% PEG5000, (+/−)-2-methyl-2,4-pentanediol (1% MPD), 1% DMSO, 1% isopropanol, 1% glycerol, 15 mM disaccharide (maltose, β-D-glucopyranosyl-D-glucose) or 1 mM different chain length fatty acids (Caprylic acid (C8), Decanoic acid (C10), Lauric acid (C12), Myristic acid (C14), Palmitic acid (C16)). The fractions were incubated in degradation buffer (25 mM Tris–HCl, PH 7.5, 150 mM NaCl) at 37 °C for 30 min. **c** ELISA assay characterizing the interactions of lipid-A with MucA$^{peri}$, MucB, MucB-NTD and mutants. lipid-A was coated in Nunc-Immuno™ MicroWell™ 96-Well Plates at a final amount of 50 μM/well. And then twofold serial dilutions of each indicated his-tag proteins were prepared. The concentration gradient is range from 100 to 0.05 μM. Following capture by lipids, the MucA$^{peri}$, MucB and MucB mutants were incubated by His-Tag antibody and horseradish peroxidase (HRP)-conjugated goat anti-mouse secondary antibody. Finally, the bound proteins were detected using TMB-ELISA substrate solution and quantified at 450 nm. Each experiment was performed three times, and each point is a mean of three replicates ± SD. **d** The sensitivity of MucB to DDM. MucB and mutants (concentration were both 130 μM) were added into degradation system and incubated at 37 °C for 30 min. The MucB-L31W mutant (130 μM) significantly decreased the sensitivity to DDM even in a long-time incubation. In the degradation system, the concentrations of MucA$^{peri}$, AlgW, and DDM are 125, 25 μM and 0.05% respectively.

under MucB in all cleavage results (Fig. 4b, d; Supplementary Fig. 5), which was proved to be the cleavage products of MucB by using mass spectrometry (MS) (Supplementary Fig. 6). This specific cleavage on MucB (the RseB is not cleaved by DegS) reflects a novel pattern in MucA/MucB system and is probably a new mechanism for *P. aeruginosa* to respond to external signal stimuli.

To validate the effects of the alkyl chain, DDM was used as the amphiphilic effector in the following MucA$^{peri}$ degradation experiments. First of all, we performed a competitive protection experiment against DDM-induced MucA$^{peri}$ cleavage in the presence of PEG (Supplementary Fig. 7). In this experiment, the PEG competed with the DDM for the binding site in MucB and the protective effect was observed to be concentration-dependent, suggesting that the PEG and amphiphilic effector binding sites are overlapped. At the same time, the lipid-ELISA assay confirmed the direct and specific binding of lipid-A to MucB but not to MucA$^{peri}$ (Fig. 4c). However, MucB-NTD exhibited lower binding to Lipid-A than that of MucB. We speculated that the CTD was necessary to relief the "plug" effect of loop$_{92-113}$ on the lipid-binding pocket. Based on our structural analysis and structural comparison (Fig. 2c), proline residues P106 and P112 are involved in the dynamic behavior of the loop$_{92-113}$. Because of the unique features, proline residues are usually important for the site-specific flexibility in protein structure. Therefore, we introduced Ala-substitutions on the two proline residues in MucB-NTD loop$_{92-113}$ to reduce its flexibility and thus to promote lipid-A binding. As we expected, the MucB-NTD-P106A and MucB-NTD-P112A mutants exhibit enhanced binding to lipid-A (Fig. 4c). Furthermore, we carried out site-directed mutagenesis on hydrophobic residues surrounding the bound PEG molecule. However, all of the Ala-substitutions in this region significantly suppressed or even abolished MucB production, indicating that the hydrophobic core is indispensable for its lipoprotein fold. Alternatively, Trp substitutions were introduced at five positions (L31W, F40W, L151W, L159W, and F176W) where the larger hydrophobic side chain would retain the protein stability but sterically hinder the interactions between amphiphilic effectors and MucB. As expected, all five Trp substitutions in MucB retained its protection effects on MucA (Supplementary Fig. 8), and the lipid-ELISA assay revealed that all of the five mutants were obviously impaired in their ability to binding lipid-A (Fig. 4c). However, the small alterations of the concave surface may not totally abolish lipoprotein function[17], accordingly, mutants F40W, L151W, L159W, and F176W still remained the sensing ability of amphiphilic effector (Fig. 4d). Even so, we identified that residue L31 was a "hot spot", in which the Trp substitution significantly decreased the sensitivity of MucB to DDM, this effect is likely attributable to its direct effects on lipid-binding and/or movements in the tunnel.

**MucA/MucB variants affect the alginate biosynthesis.** MucA and MucB are negative controllers of alginate production and their mutation results in the obvious mucoid phenotype conversion in *P. aeruginosa*. Based on the structural analysis, we further investigated the MucA mutations that disrupted the MucA$^{peri}$–MucB interactions (W158A, R162A, and H170A in MucA, Y166A was also included for comparison) or adversely affected the DDM-induced MucA$^{peri}$ degradation (L31W in MucB) in vivo to elucidate their influences on alginate biosynthesis.

None of the tested mutations affected the bacterial growth (Supplementary Fig. 9). Although both the Δ*mucA* and Δ*mucB* strains exhibited enhanced alginate biosynthesis, Δ*mucA* appeared to have more influence than Δ*mucB*. This observation highlights the profound role of MucA in suppression of AlgU

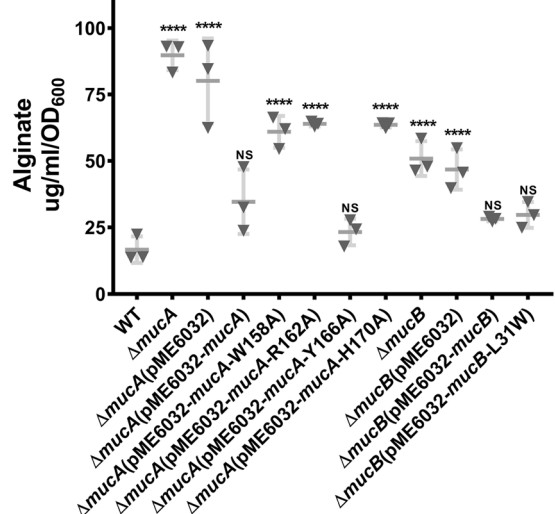

**Fig. 5 Alginate produced by *P. aeruginosa* PAO1 and its variants.** The amounts of alginate produced by PAO1, Δ*mucA*, Δ*mucB*, and various mutants were measured. Experiments were conducted in triplicates, and the error bars represent the standard deviations of the means compared to WT (PAO1). Statistical significance was calculated using a one-way ANOVA with Tukey's multiple comparison test, $P = 0.05$.

activity and suggests that MucB is the primary but perhaps not the only controlling parameter for the function of MucA (Fig. 5). As expected, mutations that attenuated the MucA–MucB association (MucAW158A, MucAR162A, and MucAH170A) increased the alginate production to a similar level as that of Δ*mucB*. In contrast, MucAY166A and MucB-L31W had no impact. In conclusion, the results from the mutagenesis and functional studies were consistent with known biological data and provide a framework to explain many mechanistic observations.

## Discussion

MucA and MucB are a pair of negative regulators of the alternative σ$^{22}$ factor AlgU and are essential for the membrane-spanning signaling in *P. aeruginosa*. This pathway controls alginate production and bacterial infection in response to extra-cytoplasmic stimuli[11]. RIP-mediated MucA degradation is a key mechanism in which accumulated misfolded protein and lipid signaling act as multiple stressors to initiate the cascading proteolytic cleavages of MucA and thus active AlgU[8]. In this work, the MucA$^{peri}$–MucB complex structure is in an inter-state revealing conformational changes preceding MucA–MucB dissociation (Fig. 1 and 2). The conserved primary binding of the MucA αI segment to the MucB C-terminal domain confirms a universal coupling pattern among their homologs in different species. The variable residues involved in the MucA/MucB interactions may provide specificity for their association (Fig. 3). We uncovered the structural basis for the LPS–MucB interaction and proved that this is a regulatory element that promotes MucA–MucB dissociation. These findings are further validated by biochemical and functional analysis (Fig. 4 and 5). These results give essential mechanistic insights into the process of LPS-induced anti-sigma factor degradation during envelope stress–response signaling in bacteria.

Protease-regulated σ$^{ECF}$ stress–response systems are widely distributed in bacteria and are involve in diverse physiological functions[18]. MucA/MucB and their *E. coli* homolog RseA/RseB represent two groups of anti-σ$^{ECF}$ factors (Fig. 3b)[12,13], bearing similar 3D structures and protein–protein binding patterns but

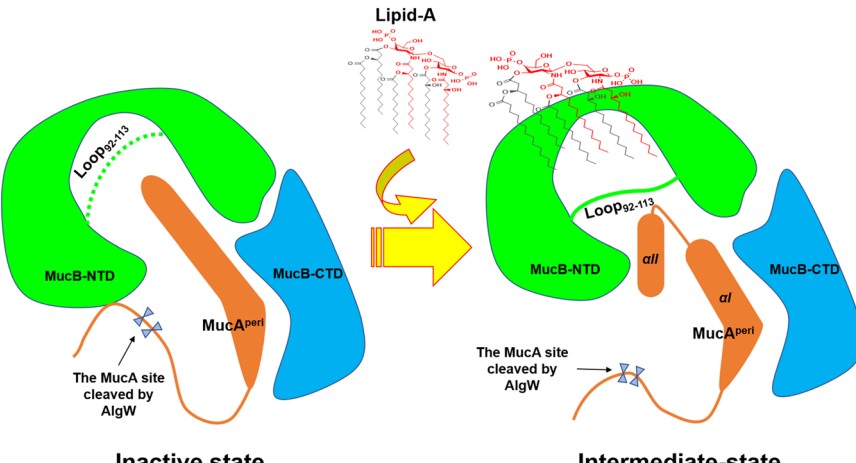

**Fig. 6 The model of MucA–MucB conformational conversions in response to lipid signals. a** Inactive state. MucB protects MucA from intramembrane proteolysis by AlgW proteases. The MucA$^{peri}$, the NTD and CTD of MucB are colored in orange, green, and blue. The loop$_{92–113}$ is represented by a green dotted line, which means that the loop is foled into the inner cavity of MucB, and forms a steric hindrance to prevent the binding of amphiphilic effectors like lipid-A. The cleavage site of MucA cleaved by AlgW is protected by MucB. **b** Intermediate state. Lipid binding induces structural changes in MucA–MucB and exposes the degradation site of MucA, these structural changes include: 1. The interaction between the NTD and CTD of MucB becomes a relatively weak van der Waals contacts mediated by water. 2. The loop$_{92–113}$ flips to the surface of MucB, forming a steric hindrance and inducing the MucA αII to insert into the inner cavity of MucB. 3. The cleavage site of MucA cleaved by AlgW was exposed to solvent, which initiates the MucA proteolysis process.

exhibiting strict interaction specificities (Supplementary Fig. 1)[7]. We speculate that the distinct binding selectivity of MucA/MucB compared with that of RseA/RseB is a consequence of the co-evolution of anti-σ$^{ECF}$ factors to allow bacteria to establish new stress-signaling pathways. Moreover, the existence of multiple types of anti-σ$^{ECF}$ factors in several bacterial species raises the possibility that selective protein–protein recognition has a role in accurately transmitting different extracellular signals[18].

Although various interactions have been observed between MucA$^{peri}$ and MucB, the intermediate binding affinity indicates that the complex is in an inter-state (Supplementary Fig. 2). This is in accordance with the feedback response mechanism used to meet the flexibility and adaptability of bacteria in response to environmental challenges. Since MucA αII does not have a significant role in MucB recognition (Fig. 3d), its insertion into the hydrophobic cavity of MucB may not function to stabilize the complex but instead serve as a "plug" that induces large conformational changes in the MucB$_{92–113}$ loop, thus ensuring efficient release of MucA cleavage site.

Because of the high structural similarity with general lipoproteins such as LppX, LolA, and LolB, the NTDs of RseB and MucB have been predicted to be the principal region responsible for binding the lipid signals[8,12,14,15]. Similar to LolB[15], the MucB-NTD in our MucA$^{peri}$–MucB structure bound with a PEG molecule in what appears to be the LPS (lipid-A) anchoring site (Fig. 4a; Supplementary Fig. 4A). The lipid-ELISA results found that the purified NTD had lower binding to lipid-A than wild-type but the Ala-substitutions on P106 or P112 exhibited significant lipid-A binding (Fig. 4c). This result validated the lipid-binding property of NTD and also revealed that the structural integrity of MucB was necessary for lipid-A binding, in which the CTD acts as a stabilizer to restrain the motion of loop$_{92–113}$.

By comparing MucB-NTD with the RseB-NTD and MucB-apo-NTD (Fig. 2a, c), we could observe dihedral angle changes in proline residues P106 and P112. The dramatic conformational variations in MucB$_{92–113}$ provide clear structural evidence to support the previous speculation that the loop connecting βE and βF serves as a lid to regulate lipophilic compound binding[14].

Consistently, the lipid-ELISA data (Fig. 4) experimentally support the "plug" mechanism of loop$_{92–113}$ to lipid-binding pocket and highlights the contribution of proline residues to this function. In addition, distinct from typical lipoproteins, this lid motion in MucB/RseB also acts as a mechanism to facilitate the release of MucA/RseA.

To characterize the nature of the effector binding at the inner cavity of the MucB-NTD, a number of organic molecules were tested for their ability to induce MucA cleavage. The results indicated that MucB appeared to interact with a variety of amphiphilic compounds implying that this system has the capability to receive even more activating signals. Our findings provide a structural template to interpret the interaction mechanism between the anti-sigma factor and lipid signals. As Fig. 6 shown, MucB–lipid-A binding results in a relatively loose packing structure of MucB and exposes the cleavage site of MucA to the solvent. The conformational change of MucB (loop$_{92–113}$) forms steric hindrance and induces the MucA αII to serve as a "plug" inserting into the inner cavity of MucB, which lead to the release of MucA from MucB. Moreover, further studies are needed to explore how MucB managed to induce local changes to facilitate access of lipophilic molecules into the hydrophobic center and whether and in what manner lipid-induced MucA–MucB structural rearrangements influence AlgW recognition.

Despite the similarities between RseA/RseB in *E. coli* and MucA/MucB in *P. aeruginosa*, it is clear that there are differences between the envelope stress responses in different bacteria because of their diverse niche adaptabilities. The distinctive structural features observed in MucB, such as the C90–C198 disulfide bond (Fig. 2a), would be of interest in future studies exploring this bacterial species-specific signaling pathway. In *P. aeruginosa*, mutation of *mucA* is a well-known mechanism for mucoid conversion and usually a major mechanism of mucoid conversion in strains isolated from CF[19]. Understanding the processes of the MucA–RIP system is important to clarify how *P. aeruginosa* is able to overproduce alginate to exhibit persistence and resistance. The specific MucA–MucB recognition event

equips a set of multiple signal-responsive elements in the MucA–RIP system. Most *mucA* mutants, which commonly result from a truncation leading to a loss of protection by MucB, achieve alginate overproduction at the cost of being insensitive to diverse envelope stresses. We expect that further investigations of the MucA–MucB-modulated signal response in *P. aeruginosa* will continue to expand our understanding of this topic and aid in the development of therapeutic strategies for treating *P. aeruginosa* infections in CF.

## Methods

**Cloning, protein expression, and purification**. The MucA[peri] (residues 106–194, UniProtKB-P38107) and the full-length MucB (UniProtKB-P38108) were amplified from the *P. aeruginosa* genomic DNA by polymerase chain reaction (PCR) using gene-specific primers (Supplementary Table 2). The genes were inserted into the plasmid (pET22b-6His) by using ClonExpress II One Step Cloning Kit (Vazyme). *E. coli* BL21 (DE3) cells (1 L), containing pET22b-MucA[peri]-6His/pET22b-MucB-6His, were cultured in Luria–Bertani (LB) medium in presence of 100 μg/mL ampicillin at 37 °C. When the $OD_{600}$ reached 0.8–1.0, protein expression was induced with 0.5 mM isopropyl-β-D-thiogalactoside (IPTG) at 16 °C for 15 h.

Bacteria were collected by centrifugation at 4000×*g* for 15 min and resuspended in 120 mL of lysis buffer consisting of 25 mM Tris–HCl (pH 7.5), 150 mM NaCl, 5% glycerol. After sonication, the supernatant was obtained by centrifugation at 15,000 × *g* for 30 min and then co-incubated with 4 mL Ni–NTA resin (Qiagen) for 1 h. The mixture was washed with lysis buffer complemented with 25 mM imidazole and target protein was eluted with lysis buffer containing 300 mM imidazole. The protein was further purified with size-exclusion chromatography Superdex™-75 (GE Healthcare), which was pre-equilibrated with solution buffer consisting of 25 mM Tris–HCl (pH 7.5), 150 mM NaCl. Peak fractions were determined by 15% SDS-PAGE gel and stain with Coomassie Brilliant Blue.

MucA[peri] and MucB were mixed at a molar ratio of 1:1 for 30 min, and the MucA[peri]–MucB complex was obtained using size-exclusion chromatography Superdex™-200 (GE Healthcare) (Supplementary Fig. 2B) in solution buffer consisting of 25 mM Tris–HCl (pH 7.5), 150 mM NaCl. Fractions containing MucA[peri]–MucB were concentrated to a concentration of approximately 15 mg/mL using a Centricon filter (30 kDa cutoff; Millipore, Billerica).

**Crystallization, data collection, and determination**. Crystallization screens were carried out by mixing protein complex with reservoir buffer at 18 °C through hanging-drop vapor diffusion method[20]. Crystals were obtained in the solution containing 0.1 M Bis-Tris (pH 6.5), 30% (v/v) Polyethylene glycol monomethyl ether 550 and 0.2 M $CaCl_2$. Crystals were soaked in cryo-protectant (reservoir solution supplemented with 20% glycerol) and flash-cooled in liquid nitrogen. Diffraction data were collected on beamline BL18U at the Shanghai Synchrotron Radiation Facility (SSRF), China. All diffraction images were processed with HKL2000 program package[21]. The structure was determined by molecular replacement using PHENIX package[22] with RseB (PDB code: 2P4B) as a template. The structure model refinement was carried out with PHENIX and COOT[23]. The final refinement statistics for the complex were summarized in Supplementary Table 1.

**Construction of *P. aeruginosa* mucA/mucB gene deletions**. A *sacB*-based two-step allelic exchange strategy was employed to construct full-length *mucA* and *mucB* deletions of *P. aeruginosa*[24]. The upstream and downstream (800 bp) PCR fragments of *mucA* and *mucB* were ligated by PCR with the gene-specific primers (Supplementary Table 2). Then target fragments were recombined to the linearized DNA fragment of pEX18Gm with ligation-free cloning system (5× Ligation-Free cloning master Mix, abm). These plasmids were transformed into *E. coli* S17-1 and then inserted into *P. aeruginosa* strains PAO1. Colonies were screened by antibiotic-resistant selection and sucrose-mediated counter-selection[20]. The *mucA* and *mucB* single-gene deletion strains were further confirmed by PCR and DNA sequencing.

**Construction of supplemented strains and mutants**. PCR-amplifed *mucA*, *mucB* and site-directed mutagenesis were cloned into the *Xho*I and *Eco*RI sites of plasmid pME6032 (Supplementary Table 2)[25]. The recombinant plasmids were transformed into corresponding gene-deleted (Δ*mucA* or Δ*mucB*) strains, which were screened by PIA plates complemented with 200 μg/mL tetracycline.

**Alginate assay**. GDP-mannuronic acid is the precursor in alginate biosynthesis, so the alginate quantification was performed using the uronic acid assay as described previously[26]. Briefly, all strains grew on PIA plates and complementation strains were induced by 0.5 mM IPTG for 36 h[27]. Cell pastes were harvested with buffer A (50 mM Tris–HCl (pH 7.4), 10 mM $MgCl_2$) and the absorbance was determined at 600 nm. Cells were removed by centrifugation at 8000×*g* for 15 min. The supernatant was added with 15 μg/mL DNase and RNase and then shaked at 37 °C for 6 h.

After nuclease digestion, proteinase K was added to a final concentration of 20 μg/mL, and the solution was incubated at 37 °C for 18 h in a shaking incubator.

For further purification, the supernatants containing dissolved alginate were placed in dialysis bags (Dialysis Membrane, Standard, RC, 10 kDa cutoff; 24 mm Width; 1.8 mL/cm, Sangon Biotech) and dialyzed in 10 mM Tris–HCl (pH 7.6) at 4 °C overnight. A 100 μL dialysis fraction was mixed with 1.0 mL borate-sulfuric acid reagent (100 mM $H_3BO_3$ dissolved in concentrated $H_2SO_4$) and 100 μL carbazole reagent (0.1% in ethanol). The mixture was heated at 55 °C for 30 min, and the absorbance was determined at 520 nm[28].

Alginate was quantified using a standard curve made from brown seaweed (BBI). Briefly, difference concentration (mg/mL) samples of commercial alginate was mixed with borate-sulfuric acid and carbazole reagent. Next, the mixture was heated at 55 °C for 30 min, and the absorbance was determined at 520 nm. The numerical values of 520 nm absorbance can be used to made a standard curve. Conversely, accompanied by the same treatment, the quantity of cell alginate was calculated through the numerical value of 520 nm absorbance of cell extractive, at the same time, the OD600 of cell culture was determined. finally, the alginate of cell culture was reported as micrograms of uronic acid per milligram of cell weight.

**Isothermal titration calorimetry (ITC)**. Isothermal titration calorimetry was used to investigate the energetics of biomolecular recognition between MucA[peri] and MucB[29,30]. All experiments were conducted in solution buffer (25 mM Tris–HCl (pH 7.5), 150 mM NaCl) at 25 °C. MucB (20 μM) was placed in the temperature-controlled sample cell and titrated with the ligand MucA[peri] (200 μM), and the total injections were made by stirring speed of 750 rpm for 19 times. Ultimately, the data were analyzed using ORIGIN software[30].

**His-tag pull-down assay**. His-tag pull-down assay was performed as described with some modifications[31,32]. First, a PreScission protease cleavage site (LEVLFQ↓GP) was inserted into the site between C terminal of MucB (or MucA[peri]) and His-tag, and the His-tag of MucB (or MucA[peri]) was removed by HRV 3C protease digestion. Then, 20 μg MucA[peri] was incubated with excessive MucB (without His-tag) and 20 μL Ni–NTA affinity resin (Qiagen) for 30 min. After centrifuging at 15,000 × *g* for 3 min, the Ni–NTA resin was washed three times with solution buffer containing 25 mM imidazole. Fractions were eluted with solution buffer containing 300 mM imidazole and determined by 15% SDS-PAGE gel and visualized by Coomassie Brilliant Blue stain.

**MucA[peri] degradation assay**. MucA[peri] was cleavaged by activated AlgW in buffer (25 mM Tris–HCl pH 7.5, 150 mM NaCl). MucA[peri] (125 μM), AlgW (25 μM), MucB (130 μM) and activation peptides (YVF or YYF, 80 μM) were incubated at 37 °C for 30 min. Variable reagents, namely 0.1 mM lipid-A hydrolyzed with NaOH[8], 0.1 mM LPS (from *E. coli* 055:B5, Solarbio) or boiled LPS (dissolve in 0.1 N NaOH aqueous solution and hydrolyze at 100 °C for 1 h, then the solution was cooled to 25 °C, and the pH was adjusted to 7), 0.05% detergents (*n*-dodecyl-β-D-Maltopyranoside, DDM; *n*-nonyl-β-D-glucopyranoside, NG; *n*-octyl-β-D-glucopyranoside, β-OG), 1% poly-ethylene glycol monomethyl ethers of different molecular weight (550, 2000, 3350, 5000), 1% organic solvents (DMSO; (+/−)-2-methyl-2,4-pentanediol, MPD; iso-propanol; glycerol), 15 mM disaccharides (maltose, β-D-glucopyranosyl-D-glucose (β-DGDG)) and 1 mM fatty acids with various chain lengths (C8–C16) were added alone in the AlgW-mediated MucA[peri] proteolysis system. The products were analyzed by SDS-PAGE and stained with Coomassie brilliant blue[33]. The DegS-mediated RseA[peri] proteolysis system was carried out as described previously[33].

**Lipid-ELISA assay**. As reported[34,35], lipid-A (sigma) were diluted in DMSO an initial amount of 50 μM/well, and the solvent (80 μL) was coated in Nunc-Immuno™ MicroWell™ 96-Well Plates (Thermo Scientific) by overnight incubation at 4 °C. Remove the solvent and blocked with a 3% (w/v) BSA in PBS for overnight at 4 °C. Thoroughly washed with PBS containing 0.05% Tween-20, and subsequently incubated with twofold serial dilutions (starting at 100 μM) of His-tagged proteins (MucA[peri], MucB, and MucB mutants) for overnight at 4 °C, the concentration gradients are 0.05, 0.1, 0.2, 0.4, 0.8, 1.6, 3.125, 6.25, 12.5, 25, 50, and 100 μM. The plates were then washed for five times with PBS, incubated with mouse anti-His antibody (Invitrogen) for 2 h, washed again for five times with PBS, incubated with horseradish peroxidase (HRP)-conjugated goat anti-mouse secondary antibody diluted 1:5000 in 1% BSA in PBS, and washed, as a final washing stage, for five times again with PBS. Finally, the bound proteins were detected using TMB-ELISA substrate solution (Horseradish Peroxidase-HRP, beyotime). After incubated for 30 min at room temperature, the reaction was stopped by addition of 1 N HCl. Optical density (OD) was subsequently measured at 450 nm. Each experiment was performed three times, and each point is a mean of three replicates ± SD.

**Statistics and reproducibility**. All experiments were performed in independent biological triplicate and the results of replicates were consistent. One-way ANOVA analyses was performed using GraphPad Prism 7 (GraphPad, CA, USA). Details of the number of biological replicates are described in the figure legends and Methods. Error bars represent standard deviation. *P* value of <0.05, which means that there is

a significant difference, P value of <0.0001 was considered as extremely significant, which is indicated with ****.

**Reporting summary**. Further information on research design is available in the Nature Research Reporting Summary linked to this article.

## Data availability

All data relevant to this study are supplied in the manuscript and supplementary files or are available from the corresponding author upon request. Atomic coordinates of the refined structures have been deposited in the Protein Data Bank (www.pdb.org) with the PDB code 6JAU.

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

## Acknowledgements

The work was financially supported by National Key Research and Development Plan under Grants 2016YFA0502700, National Natural Science Foundation of China (Grant Nos. 81871615, 81670008) and the Ministry of Science and Technology of the People's Republic of China (No. 2018ZX09201018-005). We thank National Key R&D Program of China. We thank Shanghai Synchrotron Radiation Facility (SSRF) beamline BL17U for beamtime allowance. We thank the staffs of National Center for Protein Sciences Shanghai (NCPSS) beamlines BL18U and BL19U and SSRF, Shanghai, People's Republic of China, for assistance during data collection. The search and alignment online tools used in our work were supported by NCBI (http://www.ncbi.nlm.nih.gov/). We thank Professor Yongxing He of Lanzhou University for mass spectrometry identification and analysis.

## Author contributions

B.R. designed the study and wrote this paper, L.T. completed the whole experiment and part of the article writing and revision. L.C., H.L., K.M., S.Y., S.Y., and Z.Y. participated in the X-ray data collection the structural refinement. Y.J., T.A., L.H., T.H., and W.Z. directed the revision of the content of the article, Z.N., Z.C., Y.J., H.Q., and M.X. analyzed the data. All the authors reviewed the results and approved the final version of the manuscript.

## Competing interests

The authors declare no competing interests.
