## [Peer Review file · Communications Biology]

Reviewers' comments:

Reviewer #1 (Remarks to the Author):

The first three figures of this manuscript is the recapitulation of what is already known. If the authors want to keep as part of the story, they should either streamline or put to the supplementary. These data are not novel. Pretty much the only new and interesting data is figure 4, activation of AlgW through adding lipid A, suggesting the potential binding of lipid A to MucB. This could be significant, because it is well known that that clinical mucoid isolates are associated with the loss of O antigen of LPS. That's why they are termed as "smooth" colony morphology because of the hydrophilicity of alginate they produced. The reason for that is unknown. The current paper may offer a clue to this missing puzzle. However, authors only did the experiment showing that lipid A potentially binds to MucB to relieve the inhibition to AlgW. What they should have done is the comparison of the whole LPS vs Lipid A for the AlgW efficiency, i.e, as the O antigen is being added to the growing chain of LPS, the AlgW activity is gradually decreased. That would be a lot more significant than mutational analysis to map the binding sites of lipid A. Furthermore, another key experiment, which is also missing, would be to show the direct binding between lipid A or its derivatives to the potential lipoprotein N domain of MucB through some types of labeled lipid A. This would establish the clinical significance of this study.

Reviewer #2 (Remarks to the Author):

The authors of the manuscript entitled "Molecular basis of the lipid-induced MucA-MucB dissociation in *Pseudomonas aeruginosa*" report a crystal structure of MucA-MucB complex and identified a cavity region of MucB that bound PEG molecule. The authors proposed the PEG binding site may be the site for lipid A binding and performed the proteolytic assays using AlgW, MucB and MucA. Different ligands were tested and lipid A binding residues were mutated, which confirm that lipid binding causes the dissociation of MucA and MucB. The MucA-MucB complex structure has been reported by another group and the interactions between MucA and MucB are not new. However, this manuscript revealed lipid A binding residues and proposed a mechanism that the lipid A stimulates MucB to release MucA for AlgW cleavage, which help us to understand the regulation mechanism of alginate biosynthesis.

. I have the following concerns:

- . 1. The description of the MucB has a lipoprotein-like N-terminal domain (residues 22–210) (line 110) is not accurate, as lipoprotein structures and domains are quite different. Authors should provide specific information.
- . 2. At line 411, the authors described LppX, LolA and LolB are LPS-binding proteins. Actually, they are not LPS binding proteins.

Reviewer #3 (Remarks to the Author):

The COMMSBIO-19-1548 manuscript presents the crystal structure of a MucA-MucB complex, along with biochemical and biological experiments that examined functional aspects of their findings. Based on co-crystallized PEG molecule in a hydrophobic pocket of MucB, the authors have deduced that this is a "hydrophobic pocket" at which amphiphilic ESR activating signals, such as LPS, might bind to and induce the dissociation of the MucA-MucB complex. Based on these observations, the authors made a series of MucB mutants, which convincingly demonstrate they result in the loss of Lipid-A binding to MucB. Overall, the work is novel, supports previous biochemical and biological findings, and advances the field by providing a structure that has important structure-activity relationships relevant to regulation of the bacterial ESR. In general, this reviewer found some corrections of minor importance, but several important points are made below that are relevant to a series of biochemical experiments presented. In my opinion, addressing the points raised below should not prevent publication, but they are nevertheless important points that should be addressed in a revised manuscript.

Minor points:

In Figure 2A, label the structure with an identifier for MucA-peri residues 145-158
L-124-125: Please remove "did not make contact" and "became too flexible" as the residues are not visible so nothing can be inferred. It would be helpful for readers to state the residues where MucA is cleaved are in this region, which would help support the statements in Lines 132-133.

L-126: Residues 145-158 cannot be easily identified in Figure 2A and should be labeled.

L145: clarify what is meant by "relatively loose packing structure" or remove statement

L150-157: Please label the structures with the residues being discussed in this section as they are very difficult to visualize

L157: clarify statement "induced high flexibility"

Figure 2B is difficult to interpret given that there are four colors in the superposed structures, but the legend only refers to two colors. There is a typo on the Y-axis label.

Figure 2C: this figure is not very informative as it is difficult to see where is the hydrophobic side of MucB is. It also needs a color code legend

Figure 2D: please add color code labels to the superposed image

L166: what does "loosed the interactions" mean in this context. Please provide specifics or remove.

L167-169: It is hard to visualize the regions being discussed in Figure 2D

L237: "responsible" is not the right term

L246-248: This statement is overly interpretative of the data, as binding of MucA to MucB alone could also induce stabilization of the 92-113 region

L274: The term synergy implies a specific interaction between two components of a system. Was synergy specifically tested for? Please describe how synergy was calculated or remove the term.

Line 770: The gel is mislabeled with "MucB mutant"

Important points:

1) In data presented in Figure 4B, and Supp. Figs 6 and 8, the authors present a series of biochemical experiments designed to support the binding of amphiphilic metabolites, such as LPS, to the hydrophobic cavity of MucB. This in turn relieves the inhibition of MucA degradation by AlgW, and data presented with L-IIA in Fig. 4B and Lipid-A in Fig. 4C support this model convincingly. However, the experiments performed with the non-ionic detergents DDM, BOG, and NG are difficult to interpret, given that they might disturb the interaction between MucA and MucB in a non-biologically relevant manner. This is particularly relevant under the conditions tested, since all of these detergents were used at concentrations at or higher than their CMCs in water. Therefore, the amount of monomeric DDM, BOG, and NG available in solution will be very small (or negligible, given that the great majority of the detergent molecules will be aggregated in micelles). Unfortunately, these observations also impact data presented in Supp. Fig. 8 as one possible interpretation of the DDM-PEG competition is that the increasing amounts of PEG are in fact decreasing the detergent effect of DDM on the MucA-MucB complex. How are the authors sure that the effect they are seeing is not due to a detergent effect? One suggestion is that the authors test the effect of DDM, BOG, and NG at concentrations much below their CMC. Furthermore, did the authors consider whether other detergents such as Triton X-100, lacking a long acyl chain, have a similar effect on MucA-MucB-AlgW degradation kinetics?

2) The experiments presented in Supp. Fig. 6 are also difficult to interpret, given that there is no way to confirm that the effect on relief of AlgW inhibition is due to concomitant binding of the fatty acids and the disaccharides on MucB. Furthermore, the concentrations of fatty acids (Fig. 4B, rightmost panel; Supp Fig 6) and the disaccharides are at very high, which brings into question the biological relevance of these results. Although the author's interpretation could be one possible explanation for the data presented, without further confirmation it is somewhat of a stretch to make this conclusion with these simple experiments and under these conditions.

3) In section lines 349-367, experiments are performed to examine the effect that mutation of residues important in the MucA-MucB interaction have on alginate biosynthesis. As the authors note, there are non-significant effects on alginate production. Did the authors consider that one reason why no effect was observed was because activation of the ESR pathway requires two signals? One signal to dissociate the MucA-MucB complex (as the mutations might do), and the other signal are unfolded OMP signals in the periplasmic space to activate AlgW cleavage (not induced under their experimental conditions). Therefore, under the conditions examined, is one reason why no effect was seen because there was no AlgW activation, even if the MucA-MucB complex was dissociated? The authors could conduct experiments under conditions that lead to unfolded OMP signals in the periplasm, which will better examine their model.

(All of the parts that have been revised are highlighted in green in the revised manuscript.)

Response to comments from Reviewer #1

Reviewer's comment #1: *1. The first three figures of this manuscript is the recapitulation of what is already known. If the authors want to keep as part of the story, they should either streamline or put to the supplementary. These data are not novel. Pretty much the only new and interesting data is figure 4, activation of AlgW through adding lipid A, suggesting the potential binding of lipid A to MucB. This could be significant, because it is well known that that clinical mucoid isolates are associated with the loss of O antigen of LPS. That's why they are termed as "smooth" colony morphology because of the hydrophilicity of alginate they produced. The reason for that is unknown. The current paper may offer a clue to this missing puzzle. However, authors only did the experiment showing that lipid A potentially binds to MucB to relieve the inhibition to AlgW. What they should have done is the comparison of the whole LPS vs Lipid A for the AlgW efficiency, i.e, as the O antigen is being added to the growing chain of LPS, the AlgW activity is gradually decreased. That would be a lot more significant than mutational analysis to map the binding sites of lipid A. Furthermore, another key experiment, which is also missing, would be to show the direct binding between lipid A or its derivatives to the potential lipoprotein N domain of MucB through some types of labeled lipid A. This would establish the clinical significance of this study.*

Answer: Thank you for your suggestions. All your suggestions are very important, they are of great guiding significance to my manuscript writing and scientific research work. Our statement on your concerns will be made from the following three points:

(1) According to your suggestions, we have put the original figure 1 into the supplementary material and re-numbered all the figures.

(2) As reviewer suggested, we compared the effect of LPS and lipid-A on the degradation of MucA^{peri} by AlgW (Figure 4A). As expected, both LPS and lipid-A could relief the MucB inhibition on AlgW cleavage. Although the effect of LPS is lower than that of lipid-A, the boiled LPS that release its substructure exhibits compatible effect. Thus, in lines 240, 248-251 of revised manuscript, we added the sentence "Lipid-A and LPS exhibit positive effects to relief the AlgW cleavage inhibition caused by MucB, whereas boiled LPS with enhanced core substructure accessibility could induce MucA degradation of the same magnitude as that induced by lipid-A (Fig 4B).

(3) Thank you very much for this effective and constructive comments! According to your suggestion, MucB-NTD was purified and subjected to Elisa analysis but exhibited lower binding to Lipid-A than that of MucB. We speculated that this may be due to the "plug" effect of loop₉₂₋₁₁₃ on the lipid binding pocket and CTD is necessary to relief this effect. Based on our structural analysis and structural comparison (Fig 2A), we pointed out that the dihedral angle changes in conserved P112 is important

for the dynamic behavior of the loop₉₂₋₁₁₃. Because of the unique features of proline residue (lacks an amide proton the main chain amide N is incapable of forming H-bonds), proline usually provide site-specific flexibility in structure. Therefore, we introduced Ala-substitutions on the two proline residues in MucB-NTD loop₉₂₋₁₁₃ to generate MucB-NTD-P106A and MucB-NTD-P112A. As we expected, the two MucB-NTD mutants exhibit significant binding to lipid-A (Fig 4C). The result provides another evidence to support the importance of loop₉₂₋₁₁₃ and highlights the contribution of proline residues to its function. It also revealed that the structural integrity of MucB is necessary for Lipid-A binding, in which the CTD acts as a stabilizer to restrain the motion of loop₉₂₋₁₁₃. Accordingly, we have added the above results and statements in results (lines 158, 273-282) and discussion (lines 394-401, 404-406), Figure 4C (legend: line 314), Table S2.

Response to comments from Reviewer #2

The authors of the manuscript entitled “Molecular basis of the lipid-induced MucA-MucB dissociation in Pseudomonas aeruginosa” report a crystal structure of MucA-MucB complex and identified a cavity region of MucB that bound PEG molecule. The authors proposed the PEG binding site may be the site for lipid A binding and performed the proteolytic assays using AlgW, MucB and MucA. Different ligands were tested and lipid A binding residues were mutated, which confirm that lipid binding causes the dissociation of MucA and MucB. The MucA-MucB complex structure has been reported by another group and the interactions between MucA and MucB are not new. However, this manuscript revealed lipid A binding residues and proposed a mechanism that the lipid A stimulates MucB to release MucA for AlgW cleavage, which help us to understand the regulation mechanism of alginate biosynthesis.

. I have the following concerns:

- . 1. The description of the MucB has a lipoprotein-like N-terminal domain (residues 22–210) (line 110) is not accurate, as lipoprotein structures and domains are quite different. Authors should provide specific information.*
- . 2. At line 411, the authors described LppX, LolA and LolB are LPS-binding proteins. Actually, they are not LPS binding proteins.*

Answer: Thank you for your affirmation of our work and constructive guidance. We have made the following answers of your concerns.

Reviewer’s comment #2: *1. The description of the MucB has a lipoprotein-like N-terminal domain (residues 22–210) (line 110) is not accurate, as lipoprotein structures and domains are quite different. Authors should provide specific information.*

Answer: Thank you for your suggestion. In the revised manuscript (descriptions and figures (Figure 1, 2, 4A, 6, S3)), we have revised “lipoprotein-like N-terminal domain of MucB” into “N-terminal domain of MucB (MucB-NTD)”.

Reviewer's comment #2: 2. *At line 411, the authors described LppX, LolA and LolB are LPS-binding proteins. Actually, they are not LPS binding proteins.*

Answer: Thank you for pointing out this. According to Ref 15 “as lipoproteins, both LolA and LolB have a hydrophobic cavity, which represents a possible binding site where lipoproteins may bind to lipids”, we revised these descriptions by changing “LPS-binding proteins” to “general lipoproteins” in line 390.

Response to comments from Reviewer #3

Reviewer's comment #3: *The COMMSBIO-19-1548 manuscript presents the crystal structure of a MucA-MucB complex, along with biochemical and biological experiments that examined functional aspects of their findings. Based on co-crystallized PEG molecule in a hydrophobic pocket of MucB, the authors have deduced that this is a “hydrophobic pocket” at which amphiphilic ESR activating signals, such as LPS, might bind to and induce the dissociation of the MucA-MucB complex. Based on these observations, the authors made a series of MucB mutants, which convincingly demonstrate they result in the loss of Lipid-A binding to MucB. Overall, the work is novel, supports previous biochemical and biological findings, and advances the field by providing a structure that has important structure-activity relationships relevant to regulation of the bacterial ESR. In general, this reviewer found some corrections of minor importance, but several important points are made below that are relevant to a series of biochemical experiments presented. In my opinion, addressing the points raised below should not prevent publication, but they are nevertheless important points that should be addressed in a revised manuscript.*

Answer: Thank you for your affirmation of our work and constructive guidance. We have made the following point-by-point answers to your questions.

Minor concerns:

Reviewer's comment #3: 1. *In Figure 2A, label the structure with an identifier for MucA-peri residues 145-158*

Answer: Thanks for this suggestion. In Figure 2A (We moved this figure to Figure 1 in new revision), an identifier for MucA^{peri} residues 145-158 has been added in the revised manuscript.

Reviewer's comment #3: 2. *L-124-125: Please remove “did not make contact” and “became too flexible” as the residues are not visible so nothing can be inferred. It would be helpful for readers to state the residues where MucA is cleaved are in this region, which would help support the statements in Lines 132-133.*

Answer: Thanks for this suggestion, we removed the inappropriate statement and changed this sentence to “In our MucA^{peri}-MucB complex structure, the N-terminal of MucA^{peri} (residues 106-145) is too flexible to be detected in the crystal structure” in lines 118-119.

Reviewer's comment #3: **3. L-126: Residues 145-158 cannot be easily identified in Figure 2A and should be labeled.**

Answer: As you suggested. In Figure 2A (Figure 1 in new manuscript), an identifier for MucA^{peri} residues 145-158 has been added in the revised manuscript.

Reviewer's comment #3: **4. L145: clarify what is meant by “relatively loose packing structure” or remove statement**

Answer: Comparing with the spatial arrangement of RseB N- and C-domains, the inter-domain interaction of MucB requires additional water-mediated contacts and van der Waals contacts. As reviewer concerned, since the “relatively loose packing structure” may lead to confusing the readers, we remove this statement.

Reviewer's comment #3: **5. L150-157: Please label the structures with the residues being discussed in this section as they are very difficult to visualize**

Answer: Thanks for this suggestion. The labeling on Figure 2A (Figure 1 in new manuscript) has been revised according to your comments.

Reviewer's comment #3: **6. L157: clarify statement “induced high flexibility”**

Answer: Structural comparison (Figure 1) reveals structural variations between MucB₁₉₂₋₂₁₆ and RseB₂₀₀₋₂₂₄. We speculate that besides the sequence diversity, this structural difference is also related to the distinct secondary structure elements around this region (the 6th β strand in RseB but alpha helix $\alpha 2$ in MucB). In addition, the different secondary structure adoptions may also associate with the dynamic motions of the loop₉₂₋₁₁₃. In order to make this part clear, we revised the sentence into “These distinct secondary structure elements may associate with structural variations between MucB₁₉₂₋₂₁₆ and RseB₂₀₀₋₂₂₄” in lines 149-150.

Reviewer's comment #3: **7. Figure 2B is difficult to interpret given that there are four colors in the superposed structures, but the legend only refers to two colors. There is a typo on the Y-axis label.**

Answer: Thanks, we have revised the Figure 2 and figure legend accordingly.

Reviewer's comment #3: **8. Figure 2C: this figure is not very informative as it is difficult to see where is the hydrophobic side of MucB is. It also needs a color code legend**

Answer: We regenerated figure 2C (Figure 2B in the new manuscript) to make our description of MucB hydrophobic side more visual, we add hydrophobic amino acids (gray sticks) on MucB hydrophobic side to the figure and we added relevant descriptions and color code to the legend in lines 172-174.

Reviewer's comment #3: **9. Figure 2D: please add color code labels to the superposed image**

Answer: The Figure 2D (Figure 2C in the new manuscript) had been improved according to your suggestion. The color code labels had been added to the legend,

which is described as “The loop regions (residues 92-113, residues 225-245, residues 247-275, and residues 279-291 in MucB) that generate most structural variations between MucB and MucB-apo are shown in red and magenta.” in lines 176-178. At the same time, we also marked the location of loop₉₂₋₁₁₃ and the color of the structural variations in figure 2D (Figure 2C in the new manuscript).

Reviewer’s comment #3: **10. L166: what does “loosed the interactions” mean in this context. Please provide specifics or remove.**

Answer: This sentence had been changed according to your suggestion. We have described it in detail in question 4. In order to avoid misunderstanding, we change “loosed the interactions” to “reduced the interactions” in line 159.

Reviewer’s comment #3: **11. L167-169: It is hard to visualize the regions being discussed in Figure 2D**

Answer: This error has been corrected in the revised manuscript. We marked the areas of MucB₉₂₋₁₁₃ and MucB-apo₉₂₋₁₁₃ in Figure 2D (Figure 2C in new revision).

Reviewer’s comment #3: **12. L237: “responsible” is not the right term**

Answer: According to your suggestion, we have changed this into “The hydrophobic cavity of MucB is the binding pocket for the lipid-A moiety of LPS” in lines 214-215.

Reviewer’s comment #3: **13. L246-248: This statement is overly interpretative of the data, as binding of MucA to MucB alone could also induce stabilization of the 92-113 region**

Answer: According to your suggestion, we changed this sentence into “In our PEG-bound structure, MucB₉₂₋₁₁₃ is stable and can be modeled.” in line 223.

Reviewer’s comment #3: **14. L274: The term synergy implies a specific interaction between two components of a system. Was synergy specifically tested for? Please describe how synergy was calculated or remove the term.**

Answer: Thanks for pointing this out, we removed the term “synergistic”.

Reviewer’s comment #3: **15. Line 770: The gel is mislabeled with “MucB mutant”**

Answer: This error in Supplementary Figure 8 (supplementary figure 7 in revised manuscript) has been corrected in the revised manuscript.

Major concerns:

Reviewer’s comment #3: **1. In data presented in Figure 4B, and Supp. Figs 6 and 8, the authors present a series of biochemical experiments designed to support the binding of amphiphilic metabolites, such as LPS, to the hydrophobic cavity of MucB. This in turn relieves the inhibition of MucA degradation by AlgW, and data**

presented with L-IIA in Fig. 4B and Lipid-A in Fig. 4C support this model convincingly. However, the experiments performed with the non-ionic detergents DDM, BOG, and NG are difficult to interpret, given that they might disturb the interaction between MucA and MucB in a non-biologically relevant manner. This is particularly relevant under the conditions tested, since all of these detergents were used at concentrations at or higher than their CMCs in water. Therefore, the amount of monomeric DDM, BOG, and NG available in solution will be very small (or negligible, given that the great majority of the detergent molecules will be aggregated in micelles. Unfortunately, these observations also impact data presented in Supp. Fig. 8 as one possible interpretation of the DDM-PEG competition is that the increasing amounts of PEG are in fact decreasing the detergent effect of DDM on the MucA-MucB complex. How are the authors sure that the effect they are seeing is not due to a detergent effect? One suggestion is that the authors test the effect of DDM, BOG, and NG at concentrations much below their CMC. Furthermore, did the authors consider whether other detergents such as Triton X-100, lacking a long acyl chain, have a similar effect on MucA-MucB-AlgW degradation kinetics?

Answer: Really thank you for your positive comments and valuable suggestions.

(1) In this manuscript, we use nonionic detergents to simulate the effect of lipid-A on the degradation of MucA because they are molecules with hydrophilic headgroup and aliphatic side chains, possessing similar structure with L-IIA (the proposed minimal LPS fragments bind RseB or MucB, Ref 8). As you mentioned, the concentration of Nonionic detergents used in our experiment is higher than their CMCs in water, so we repeat the degradation experiments in which the concentration of Nonionic detergents is lower than CMCs (Supplementary Fig 5A in the revised manuscript). The results show that, even under the CMCs, nonionic detergents still have the same effect as lipid-A. In addition, we also explored other nonionic detergents Triton X-100, and the results show that they have a similar effect on the degradation of MucA, as expected, the effect is weaker than that of detergents with long acyl chains. In lines 252-253 of the revised manuscript, we added a description of the result.

(2) For Supplementary Fig 8 (Supplementary Fig 7 in the new manuscript), we added negative controls (the last two columns). 15% glycerol did not inhibit the detergent-caused MucA-MucB dissociation and subsequent AlgW digestion on MucA, indicating that the effect of PEG is at least not because of the altered viscosity of the solution or diffusing capacity of the detergent. In addition, 15% PEG does not affect the degradation of MucA by AlgW (the last column), indicating that the effect of PEG is also not due to the direct interactions to MucA and AlgW.

Reviewer's comment #3: *2. The experiments presented in Supp. Fig. 6 are also difficult to interpret, given that there is no way to confirm that the effect on relief of AlgW inhibition is due to concomitant binding of the fatty acids and the dissacharides on MucB. Furthermore, the concentrations of fatty acids (Fig. 4B, rightmost panel; Supp Fig 6) and the dissacharides are at very high, which brings*

into question the biological relevance of these results. Although the author's interpretation could be one possible explanation for the data presented, without further confirmation it is somewhat of a stretch to make this conclusion with these simple experiments and under these conditions.

Answer: Thank you very much for asking this question. Indeed, as you pointed out, our result demonstrates the additive effects of the fatty acids and disaccharides but further investigations are required to prove the concomitant binding of the two effectors on MucB. We agree that the current data is inadequate to suggest the idea that the fatty acids and disaccharides have biological function to trigger the MucA-MucB dissociation in bacterial cell, but the result indicated that the hydrophilic headgroup and aliphatic side chain were both functional groups of lipid-A for RIP pathway signaling. In addition, it provides a clue to investigate whether there are other biological amphiphilic effectors could be involved in RIP signaling. Therefore, we revised the related statement to “These data suggest that an alkyl chain with a hydrophilic headgroup are both functional groups of lipid-A for inducing MucA^{peri} release.” in lines 256-257.

Reviewer's comment #3: *3. In section lines 349-367, experiments are performed to examine the effect that mutation of residues important in the MucA-MucB interaction have on alginate biosynthesis. As the authors note, there are non-significant effects on alginate production. Did the authors consider that one reason why no effect was observed was because activation of the ESR pathway requires two signals? One signal to dissociate the MucA-MucB complex (as the mutations might do), and the other signal are unfolded OMP signals in the periplasmic space to activate AlgW cleavage (not induced under their experimental conditions). Therefore, under the conditions examined, is one reason why no effect was seen because there was no AlgW activation, even if the MucA-MucB complex was dissociated? The authors could conduct experiments under conditions that lead to unfolded OMP signals in the periplasm, which will better examine their model.*

Answer: That's a good question. As you mentioned, activating the ESR pathway requires dual signals for MucA-MucB dissociation and AlgW activation. MucBL31W which disturbed the lipid binding ability of MucB was used as a mutant with impaired sensitivity to lipid signals, but it did not exhibit obvious changes in alginate production. It is possible due to that the RIP system is normally maintained in off-state and requires dual signals to switch to on-state (Ref 8), the effect of MucBL31W could be detected under conditions when the RIP is turned on by the dual signals. We agree that a suitable and stable lipids induction model could further support our results. Actually, we did try to establish the *Pseudomonas aeruginosa* mucus transformation model by using ammonium metavanadate to induce palmitate modification of lipid-a (Damron F H, Davis Jr M R, Withers T R, et al. Vanadate and triclosan synergistically induce alginate production by *Pseudomonas aeruginosa* strain PAO1[J]. *Molecular microbiology*, 2011, 81(2): 554-570.), and to use the d-cycloserine lactone to induce the high expression of algD operon, which is related to the secretion of alginate and the transformation of mucus type (Wood L F, Leech A

J, Ohman D E. Cell wall- inhibitory antibiotics activate the alginate biosynthesis operon in Pseudomonas aeruginosa: roles of σ 22 (AlgT) and the AlgW and Prc proteases[J]. Molecular microbiology, 2006, 62(2): 412-426.). So far, we have not established a stable system yet. We will devote more energy on this in our further studies.

Once again, we sincerely thank the editor and the three reviewers for their careful reading and valuable advice of our manuscript. Thank you very much for all your time and effort.

REVIEWERS' COMMENTS:

Reviewer #1 (Remarks to the Author):

There was no specific information provided about the source of LPS. Is this the whole LPS such as those from nonmuroid reference strain PAO1, or LPS without a long O antigen side chain such as those of an isogenic *muA* mutant? Is this LPS commercially available? Also, why choose 0.1 mM lipid A? is this a physiologically relevant concentration? What is the concentration of LPS used in Figure 4, same as 0.1 mM? How the boiled LPS was prepared? Generally speaking, LPS is very stable, and resistant to heat treatment. That's why we use a method called hot phenol-water extraction. How do you know that the boiled LPS structure is truly exposed?

Reviewer #2 (Remarks to the Author):

The authors have satisfactorily addressed all of my concerns and I recommend the article be accepted for publication.

(All of the parts that have been revised are highlighted in **green** in the revised manuscript.)

Response to comments from Reviewer #1

Reviewer's comment #1: *There was no specific information provided about the source of LPS. Is this the whole LPS such as those from nonmucoid reference strain PAO1, or LPS without a long O antigen side chain such as those of an isogenic mucA mutant? Is this LPS commercially available? Also, why choose 0.1 mM lipid A? is this a physiologically relevant concentration? What is the concentration of LPS used in Figure 4, same as 0.1 mM? How the boiled LPS was prepared? Generally speaking, LPS is very stable, and resistant to heat treatment. That's why we use a method called hot phenol-water extraction. How do you know that the boiled LPS structure is truly exposed?*

Answer: Thank you for your insights and questions. According to reference 8 (Santiago Lima *et.al.* Science, 2013), >0.1 mM of LPS could displace RseA from RseB, they also found that the active structure L-IIA (obtained by boiling LPS under NaOH solution) can antagonize both RseA-RseB and MucA-MucB binding. So we purchased the Lipopolysaccharides (LPS) from Solarbio (Lot L8880, purified from *Escherichia coli* 055:B5 by phenol extraction), and use 0.1 mM of Lipid-A in our experiments including Figure 4. We speculated that the over dose of LPS and lipid-A used in in-vitro binding assay could simulate the outer membrane damage effect which activated the RIP pathway. The boiled LPS was dissolved in 0.1N NaOH and was neutralized to pH 7, according to reference 8, LPS could be hydrolyzed into L-IIA under strong alkali and high temperature environment. In the returned manuscript, we added information about the source of LPS and the preparation process for boiled LPS in lines 199, 465-468 and 574. And as you concerned, we revised the statements as “whereas boiled LPS under alkali solution with activate fragment L-IIA released could induce...” in lines 208-210.

Once again, we sincerely thank the editor and the three reviewers for their careful reading and valuable advice of our manuscript. Thank you very much for all your time and effort.